# Identification of Key Elements in Prostate Cancer for Ontology Building via a Multidisciplinary Consensus Agreement

**DOI:** 10.3390/cancers15123121

**Published:** 2023-06-08

**Authors:** Amy Moreno, Abhishek A. Solanki, Tianlin Xu, Ruitao Lin, Jatinder Palta, Emily Daugherty, David Hong, Julian Hong, Sophia C. Kamran, Evangelia Katsoulakis, Kristy Brock, Mary Feng, Clifton Fuller, Charles Mayo, BDSC Prostate Cancer Consortium

**Affiliations:** 1Department of Radiation Oncology, University of Texas MD Anderson Cancer Center, Houston, TX 77030, USA; cdfuller@mdanderson.org; 2Department of Radiation Oncology, Loyola University Medical Center, Berwyn, IL 60402, USA; asolanki@luc.edu; 3Department of Biostatistics, University of Texas MD Anderson Cancer Center, Houston, TX 77030, USA; txu2@mdanderson.org (T.X.); rlin@mdanderson.org (R.L.); 4Department of Medical Physics, Virginia Commonwealth University, Richmond, VA 23284, USA; jatinder.palta@vcuhealth.org; 5Department of Radiation Oncology, University of Cincinnati College of Medicine, Cincinnati, OH 45267, USA; daugheec@ucmail.uc.edu; 6Department of Radiation Oncology, University of Southern California, Los Angeles, CA 90089, USA; david.hong@med.usc.edu; 7Department of Radiation Oncology, University of California San Francisco, San Francisco, CA 93701, USA; julian.hong@ucsf.edu (J.H.); mary.feng@ucsf.edu (M.F.); 8Department of Radiation Oncology, Massachusetts General Hospital, Boston, MA 02129, USA; sophia_kamran@post.harvard.edu; 9Department of Radiation Oncology, James A Haley VA Medical Center, Tampa, FL 33612, USA; ekatsoulakis@usf.edu; 10Department of Imaging Physics, University of Texas MD Anderson Cancer Center, Houston, TX 77030, USA; kkbrock@mdanderson.org; 11Department of Radiation Physics, University of Michigan, Ann Arbor, MI 48109, USA; cmayo@med.umich.edu

**Keywords:** prostate cancer, clinical guidelines, treatment-related toxicities, informatics, ontology

## Abstract

**Simple Summary:**

Prostate cancer (PCa) is one of the most common cancers and the second leading cause of cancer-related deaths in men in the United States. Accurate diagnosis, management, and posttherapy surveillance are critical for optimizing survival and patient quality of life. Sharing information among providers is a challenge due to the use of different health information systems or data capture workflows which can lead to ambiguity and misinterpretation of shared information. The aim of our study was to formulate an expert panel-based consensus on PCa-specific key data elements. Using the Delphi method, PCa experts developed a two-tiered thirty-item list of treatment-related toxicities for standardized clinical data capture. Additionally, four multi-domain symptom questionnaires were ranked, and definitions on disease control metrics were formalized. These findings have been used to develop a comprehensive operational ontology for PCa care that can facilitate knowledge sharing and scalable machine learning approaches.

**Abstract:**

Background: Clinical data collection related to prostate cancer (PCa) care is often unstructured or heterogeneous among providers, resulting in a high risk for ambiguity in its meaning when sharing or analyzing data. Ontologies, which are shareable formal (i.e., computable) representations of knowledge, can address these challenges by enabling machine-readable semantic interoperability. The purpose of this study was to identify PCa-specific key data elements (KDEs) for standardization in clinic and research. Methods: A modified Delphi method using iterative online surveys was performed to report a consensus agreement on KDEs by a multidisciplinary panel of 39 PCa specialists. Data elements were divided into three themes in PCa and included (1) treatment-related toxicities (TRT), (2) patient-reported outcome measures (PROM), and (3) disease control metrics (DCM). Results: The panel reached consensus on a thirty-item, two-tiered list of KDEs focusing mainly on urinary and rectal symptoms. The Expanded Prostate Cancer Index Composite (EPIC-26) questionnaire was considered most robust for PROM multi-domain monitoring, and granular KDEs were defined for DCM. Conclusions: This expert consensus on PCa-specific KDEs has served as a foundation for a professional society-endorsed, publicly available operational ontology developed by the American Association of Physicists in Medicine (AAPM) Big Data Sub Committee (BDSC).

## 1. Introduction

Prostate cancer (PCa) is the most diagnosed cancer and the second leading cause of cancer-related deaths among men in the United States, requiring comprehensive multidisciplinary assessment and personalized management [1,2]. Depending on patient preference, comorbidities, and extent of disease at diagnosis, PCa patients may be eligible for continually evolving local therapy options including radical prostatectomy, low-dose or high-dose-rate brachytherapy, and external beam radiotherapy delivered using photons and/or protons [3,4,5,6,7]. Clinicians are strongly encouraged to provide their PCa patients with individualized risk estimates of post-therapy toxicities and PCa recurrence [8], and to clearly communicate cancer management and outcomes with other multidisciplinary providers. However, these critical goals are becoming increasingly difficult to perform given the exponential rates of electronic data growth, inconsistent data reporting, and the use of ambiguous language in clinical documentation. For example, definitions for locoregional recurrence after radiotherapy in PCa can vary among clinicians (i.e., pelvic and/or aortic nodal regions). Even semantic (i.e., meaning) ambiguity in reporting properties of ‘active surveillance’ versus ‘watchful waiting’ in men with localized PCa was identified as a significant issue in another consensus study [9]. Without expert-endorsed identification and standardization of key data elements (KDEs) related to PCa care, the interpretation and reuse of such data (i.e., advanced analytical techniques, such as machine learning approaches) and interoperable data exchange will continue to be significantly hindered.

Lack of disease-specific KDE definitions is a major barrier to improving research and clinical decision-making processes. A systematic review of 17 consensus-forming studies aimed at identifying clinical outcomes for standardization in general cancer research found that nearly 60% of the studies lacked specific recommendations on measuring or defining KDEs, including those related to posttherapy patient-reported symptoms [10]. Already published consensus-forming studies on PCa care and/or reporting guidelines have focused on elements related to subpopulations of PCa patients (i.e., patients with oligometastatic disease) or defined a narrow scope of clinical care metrics without an explicit nomenclature for specifying standardized value sets [11,12,13,14,15]. Even when KDEs are identified, data are often siloed in different information systems or entered in electronic health record (EHR) systems using free text entries which results in lower data quality and higher risks of semantic ambiguity [16]. The extensive heterogeneity existing in electronic data source-based file formatting and data structuring (i.e., schemas) prevents effective search, retrieval, pooling, and analysis of complex biomedical data [17,18,19]. Therefore, there is a significant need for ongoing data standardization of KDEs, preferably expressed through an ontology, for uniform reporting across all PCa patients.

In 1998, Studer et al. defined an ontology as ‘a formal, explicit specification of a shared conceptualization’ [20]. In other words, ontologies are a consensus-based, machine-readable representation of knowledge with explicit definitions for concepts and the relationships between them. Operational ontologies build upon ‘upper level’ ontologies, such as the Open Biological and Biomedical Ontologies (OBO) Foundry [21], and address operational gaps to provide standardized value sets for sub-domains of knowledge. Once implemented, highly expressive ontologies can vastly improve patient care and cancer research endeavors by preserving semantics during information retrieval and knowledge sharing, enabling knowledge inference, and facilitating the creation of large databases for advanced analytics [22,23]. Despite the numerous advantages associated with ontologies, limited work has been published on the translation of expert-guided guidelines for building publicly available operational ontologies.

The primary objective of this study therefore was to reach a multidisciplinary, expert-based consensus on KDEs requiring standardization in PCa care. Herein, we report the results of a modified Delphi procedure used to establish universal PCa-specific KDEs associated with three major concepts of care: (1) treatment-related toxicities (TRTs), (2) patient-reported outcome measures (PROMs), and (3) disease control metrics (DCMs). This study addresses a critical gap in existing knowledge by providing a tiered list of TRTs, a ranking of PROM tools, and explicit definitions for DCMs. Clinical applications and future directions are also provided. To our knowledge, this is the first expert Delphi consensus study in cancer to be directly applied to the development of an operational ontology.

## 2. Materials and Methods

American Association of Physicists in Medicine—Big Data Sub Committee. The American Association of Physicists in Medicine (AAPM) is a scientific and professional organization with extensive expertise in improving treatment accuracy and patient safety through the development and implementation of consensus-driven information standards, such as the Standardizing Nomenclatures in Radiation Oncology Report (TG-263) [24,25]. Chartered in 2019, AAPM’s Big Data Sub Committee (AAPM BDSC) is a multi-institutional committee composed of a diverse group of stakeholders including physicians, medical physicists, and informaticians who are also representatives of professional organizations, such as the American Society of Radiation Oncology (ASTRO), the European Society of Therapeutic Radiation Oncology (ESTRO), the Canadian Organization of Medical Physicists (COMP), and NRG Oncology [26]. Members of the BDSC have significant experience in data standardization and clinical/research needs assessments, making the BDSC a suitable expert working group to develop the operational ontology for oncology (O3) for PCa. To more accurately build and express core knowledge concepts in the ontology to be used by PCa specialists, the BDSC created the BDSC Prostate Cancer Consortium, otherwise known as the Expert Panel, to participate in this modified Delphi consensus initiative.

Expert Panel. A list of nationally recognized multidisciplinary specialists located in the United States with domain expertise in the management of genitourinary (GU) malignancies (*n* = 87) was thoughtfully drafted through recommendations from members of the BDSC and ASTRO. Personalized emails were sent to each expert with a two-week response deadline; 48 experts agreed to participate. Throughout the Delphi process, 39 participants (the Expert Panel) completed at least one remote electronic survey. Panel characteristics are reported in Table 1, with diverse representatives from radiation oncology, radiation physics, medical oncology, and urology. All clinical experts were known to have extensive practice in the management of PCa. Members of the Expert Panel and Delphi biostatisticians are listed in Box 1.

Box 1Members of the BDSC Prostate Cancer ConsortiumDelphi Consensus Lead: Amy Moreno (MD Anderson Cancer Center)Expert Panel: Neeraj Agarwal (University of Utah Huntsman Cancer Institute), Ana Aparicio (MD Anderson Cancer Center), Jeffrey Cadeddu (University of Texas Southwestern Medical Center), Ronald Chen (University of Kansas Medical Center), Seungtaek Choi (MD Anderson Cancer Center), Matthew Cooperberg (University of California San Francisco), Alan Dal Pra (University of Miami Miller School of Medicine), Indra Das (Northwestern University Feinberg School of Medicine), Neil Desai (University of Texas Southwestern Medical Center), Dayssy Diaz Pardo (Ohio State University Comprehensive Care Center), Weiliang Du (MD Anderson Cancer Center), William Hall (Medical College of Wisconsin), Celestia Higano (University of Washington, Fred Hutchison Cancer Research Center), Karen Hoffman (MD Anderson Cancer Center), Maha Hussain (Northwestern University Feinberg School of Medicine), Sophia Kamran (Massachusetts General Hospital Cancer Center, Harvard Medical School), Amar Kishan (University of Cali-fornia Los Angeles), Bridget Koontz (East Carolina University), Rajat Kudchadker (MD Anderson Cancer Center), Charles Mayo (University of Michigan), Jeff Michalski (Washington University School of Medicine), Alicia Morgans (Northwestern University Feinberg School of Medicine), Himanshu Nagar (Weill Cornell Medicine), Louis Potter (Northwell Health, Zucker School of Medicine), Tyler Robin (University of Colorado Denver School of Medicine), Mihaela Rosu-Bubulac (Virginia Commonwealth University), Howard Sandler (Cedars-Sinai), Neal Shore (Atlantic Urology Clinics), Abhishek Solanki (Loyola Medicine), Cora Sternberg (Weill Cornell Medicine), Rahul Tendulkar (Cleveland Clinic), Ying Xiao (University of Pennsylvania, Perelman School of Medicine), James Yu (Trinity Health of New England), Zachary Zumsteg (Cedars-Sinai), and unnamed panelists (*n* = 5).Delphi Statisticians: Ruitao Lin (MD Anderson Cancer Center), and Tianlin Xu (MD Anderson Cancer Center)

Modified Delphi Method. We used a fully remote, two-step modified Delphi technique to formulate the expert consensus on KDEs relevant to PCa care. The Delphi method, initially developed in the 1950s, has been widely adopted and adapted for its utility in information gathering and establishing consensus on policies or decision-making processes [27]. Controlled feedback and anonymity are two strengths of this research approach as experts reply anonymously to a series of questionnaires and can adjust their opinions after reviewing interval feedback or insights from the entire group [28]. Unlike traditional Delphi, the modified Delphi is designed to allow researchers to carefully research and create a curated list of initial items or questions for review by the panel to facilitate the first round of the consensus-seeking process [29]. Panel sizes can vary significantly from 3 to over 70 members [28], therefore, it is advised to have at least one representative from each stakeholder group to ensure that substantive sub-domain knowledge is integrated and communicated throughout the entire process. This modified Delphi was approved by The University of Texas MD Anderson Cancer Center Institutional Review Board.

Survey Design and Administration. We performed an informal but comprehensive literature review to develop a list of potentially important data elements related to PCa. Specifically, PubMed was iteratively queried using a variety of combinations of the following keywords and/or MeSH terms: prostate cancer, guidelines, treatment-related toxicities, adverse effects, disease control metrics, disease response assessment, prostate cancer surveillance, patient-reported outcomes, symptom monitoring, PRO tools and/or validated questionnaires. Abstracts were reviewed for relevance and manuscript review was limited to publications related to large PCa randomized clinical trials and/or PCa guidelines endorsed by nationally/internationally recognized organizations, such as the American Urological Association (AUA), ASTRO, the European Society for Therapeutic Radiation and Oncology (ESTRO), and the European Society of Urogenital Radiology (ESUR). Based on the review of the literature and serial BDSC meetings with GU specialists, data elements were categorized into three overarching concepts that included TRTs (*n* = 43), PROMs (1 urinary, 1 sexual health, 1 erectile dysfunction, and 4 validated multi-domain questionnaires), and DCMs (open-ended questions). Two rounds of electronic surveys were conducted between October 2020 to February 2021. Experts were given approximately 3 weeks to complete the first survey with several automated reminder emails delivered. Analysis of results from the first questionnaire was performed during the holiday-filled months of November–December and included a manual review and categorization of open-ended questions and new ones submitted by the experts. The second survey was deployed in January and contained anonymous group feedback for review and summarization of final KDEs. After completion of the Delphi consensus process, the BDSC began to work on designing an operational ontology based on the Expert Panel’s recommendations.

Treatment-Related Toxicities. During Round 1, experts were asked to rate 43 CTCAE V5.0, toxicities on a 9-point numerical scale (1 = not important to 9 = very important). This scale was intentionally selected based on prior publications demonstrating unique advantages for granular analysis and easy conversion to a 3-point-based scale [30]. During Round 2, panelists were presented with statistical representations of group responses from Round 1 (see Statistical Analysis below). Based on their own interpretations of the group’s insight, each panelist was then tasked with categorizing toxicity metrics into three tiers: (1) a “minimum required” or tier 1 element list, (2) a “strongly encouraged but not mandatory” or tier 2 list, and (3) a “not required” list. The first tier was defined as a list of critical data elements that should be mandatory for standardized reporting on a national level to ensure equivalent high-quality data exchange and analysis for all PCa patients. The second-tier list was intentionally defined more broadly to include additional metrics that were recommended by the panel for standardized collection, if possible, in an effort to facilitate the development of a highly expressive operational ontology by the BDSC. The third list included all the remaining toxicities that the experts considered to be of least importance to prevent fatigue.

Patient-Reported Outcome Metrics. After answering questions about TRTs during the first survey, experts were asked to score the following validated PROM tools: the International Prostate Symptom Score (IPSS, for evaluation of severity of urinary symptoms and impact on quality of life) [31] the International Index of Erectile Function (IIEF, for evaluation of erectile dysfunction) [32] and the Sexual Health Inventory for Men (SHIM, for evaluation of sexual health and erectile dysfunction) [33]. Additionally, four commonly used PCa-specific, multi-domain PROM questionnaires were reviewed for their usage and ability to adequately assess urinary, sexual, bowel, and systemic functions as well as quality of life. These included the Expanded Prostate Index Composite-26 (EPIC-26), the European Organization for Research and Treatment of Cancer Quality of Life Questionnaire-Prostate Cancer module (EORTC QLQ-PR25), the Functional Assessment for Cancer Therapy Prostate Cancer module (FACT-P), and the Patient-Reported Outcomes Common Terminology Criteria for Adverse Events (PRO-CTCAE) [34,35]. Since consensus was achieved on the PROM questionnaire rankings during Round 1, no further related questions were asked during the second survey.

Disease Control Metrics. Potential DCM metrics, which are numerous and have greater variability in existing definitions (i.e., the definition of biochemical failure and/or oligometastatic PCa), were open to discussion and item list generation during Round 1. DCM-related questions were intentionally open-ended for comprehensive information gathering from the experts and for qualitative analysis. During Round 2, further refinement of the definitions for biochemical recurrence and oligometastatic disease, as well as the development of a list of recurrent-specific KDEs for standardized reporting was completed by the panel. Figure 1 shows the overall study process flowchart for consensus formation on PCa-related KDEs with translation to create O3.

Statistical Analysis. Descriptive statistics, including central tendencies (i.e., means), counts, and percentages, were performed to describe Expert Panel characteristics and KDE variables according to their data types. Variables that were scored from 1 to 9 during Round 1 were additionally categorized into three agreement levels during Round 2. Scores ranging from 7 to 9 were coded as ‘high’ in agreement, 4 to 6 as ‘medium’, and 1 to 3 as ‘low’ in agreement. Individual item-based agreement indices (AI) were also calculated after each round to aid in statistical ranking. Agreement indices measure the percentage of equal ratings for the question, *I* (in round t), over a specific number of panelists who responded to the survey, where 0 < AI < 1 (0 denotes total disagreement, and 1 denotes consensus) [36]. To reflect the correlation within each question in Round 2, the intraclass correlation coefficient (ICC) was calculated using a linear mixed model with random effects for TRT-related questions. The ICC was interpreted as follows: excellent (>0.75), fair to good (0.40–0.75), and poor (<0.40) [37]. The chi-square test or Fisher’s exact test (when any category had <5 observations) was used to test whether there was a difference in the panel categorization of TRT items between Rounds 1 and 2. Consensus was considered to be reached when percent scoring and AI per question reached above 50%. Metrics and thresholds for indicating consensus are highly flexible in Delphi studies and include meeting a percent threshold, using standard deviations, or interquartile range constraints [28,38,39]. We selected a lower agreement percentage threshold in combination with the agreement index to account for variability in panel response size per round and to allow for greater inclusion of data elements into our second-tier TRT list. This consensus process was communicated with the Expert Panel and both the panel and BDSC had to review and approve the consensus recommendations on PCa-specific KDEs listed herein to proceed with the development of the operational ontology. Calculations were performed using the R and package lme4 [40].

## 3. Results

### 3.1. Panel Characteristics and Participation

Characteristics of the multidisciplinary Expert Panel are presented in Table 1. Two-thirds of the participants were male, 56% were radiation oncologists, and 79% reported practicing in an academic setting. The average number of years in practice and weekly patient caseloads were 18 years and 30 patients, respectively. Of 39 panelists who completed at least one survey, 34 (87%) completed Round 1, 26 (67%) completed Round 2, and 25 (64%) completed both surveys.

### 3.2. Treatment-Related Toxicity KDEs

Forty-three TRTs were reviewed and scored by the experts, including items related to rectal (i.e., rectal hemorrhage, and rectal fistula), urinary (i.e., urinary incontinence, and urinary frequency), hormonal (i.e., hot flashes, and libido decrease), and constitutional (i.e., fatigue) symptoms. Consensus on the first-tier (minimum KDE), second-tier (strongly encouraged), and third-tier (not required) lists are shown in Table 2 along with corresponding agreement indices. The first tier consisted of 15 toxicities that achieved ≥50% selection for this list and ≥50% calculated agreement index. The second tier included another 15 variables, of which 10 (67%) had ≥50% selection for this list with an agreement index range of 0.35 to 0.49. The remaining five variables met one of the above requirements or were added to this list owing to exclusion from the “not required” list. Twelve remaining TRTs were ranked as low yield for reporting. No additional TRT-related items were recommended for review by the panel.

In general, the panel demonstrated high rates of agreement (i.e., high AI) when reviewing TRTs that were of very high or low importance. Those with higher AI variability (i.e., greater disagreement among panelists) tended to be categorized in the second-tier list. The ICC for TRTs was 0.58, and the chi-square/Fisher’s exact test showed a significant difference in panelist answers between both rounds of surveys for 23 out of the 43 TRTs.

### 3.3. Patient-Reported Outcome Metrics KDEs

The percentage of panelists reporting ‘almost always’ use of the IPSS, SHIM, and IIEF questionnaires was 56%, 29%, and 6%, respectively. When asked whether panelists preferred the SHIM or IIEF tool, 53% preferred SHIM, 32% preferred IIEF, and 15% preferred neither. Panelists who reported “always” using the EPIC-26, EORTC QLQ-PR25, FACT-P, and PRO-CTCAE multi-domain questionnaires represented 24%, 6%, 0%, and 0% of the group, respectively. Nearly two-thirds of the panel reported never using the FACT-P and PRO-CTCAE tools. Overall, EPIC-26 scored the highest in all domains with an average multi-domain score of 7.5 out of 9 and an agreement index of 0.77 (Figure 2). This was consistent with the panel’s final ranking of the PROM tools: EPIC-26 (highest ranking), EORTC QLQ-PR25, FACT-P, and PRO-CTCAE (lowest ranking).

### 3.4. Disease Control Metrics KDEs

The panel offered several recommendations for reporting locoregional recurrences. They included a high preference (61%) for reporting detailed documentation on primary site recurrences (i.e., localized in the prostate, seminal vesicles, or prostate bed) and granular documentation on nodal recurrences (i.e., reporting recurrences as “none”, “pelvic”, “aortic”, and/or “distant”). After reviewing the open-ended questions from Round 1, panelists were presented with a list of DCM options during Round 2. Overall ratings and associated agreement indices are shown in Table 3. Using a similar tier categorization as performed for TRTs, 11 out of 14 disease status elements were identified as DCM-specific KDEs by the multidisciplinary panel. “Stable disease”, “Indeterminate (possible pseudo-progression)”, and “Partial Response” were not prioritized as optimal DCMs due to their vague definitions in comparison to the prioritized terms.

Regarding the extent of metastatic disease, several definitions of metastatic burden were reviewed, including the CHAARTED and LATITUDE criteria from randomized clinical trials [41,42]. Based on clinical utility, the panel recommended standardizing the reporting of osseous metastatic lesions using a numerical value up to five and reporting visceral (i.e., organ) metastatic lesions in a binary format (i.e., either present or absent).

## 4. Discussion

This Delphi method highlights ongoing and significant heterogeneity in how providers manage and report clinical data on PCa patients. By the end of the study, 15 TRTs were identified by the Expert Panel as having high importance in assessing outcomes related to PCa (tier 1), and another 15 were considered to be of moderate value (tier 2) for enhancing the expressivity of a PCa-specific operational ontology [43]. This component of the study is unique as it provides a statistically tiered list approach for meaningful KDEs that are provider-based (i.e., objective) in addition to the corresponding patient-reported symptoms during and after therapy (i.e., subjective). The prioritized lists inform resource investment directions within organizations and allow for greater standardization and expressivity in an operational ontology.

### 4.1. Recommended Treatment-Related Toxicity Reporting

Panel recommendations for key TRTs to report focused mainly on urinary and bowel-related symptoms. Rectal hemorrhage, urinary incontinence, and urinary retention, all of which had the highest agreement indices, are known to be common acute and chronic treatment-related complications after radiation therapy or surgery [43]. For patients treated with radiotherapy, there is substantial evidence to support the radiation dose-volume effects in radiation-induced rectal and/or urinary injury [44,45]. This has led to the development of treatment planning guidelines with dose volume constraints, such as those outlined by QUANTEC, and the investigation of endorectal balloons and rectal spacers as methods to further reduce risks of rectal injury [45,46,47,48]. Moreover, large randomized clinical trials investigating the efficacy of different regimens (i.e., hypofractionation and/or stereotactic body radiation therapy) also serve as additional references to clinicians to mitigate TRT risks using schedule-dependent dose-volume constraints [3,4,49,50]. Prospective and standardized reporting of TRT-KDEs in clinical practice is critical to validate the long-term outcomes of therapy beyond the time window reported in clinical trials. One method of supporting such data collection is through the use of ‘smart forms’ or structured clinical note templates in the patient’s electronic health record chart. Problem-oriented templates provide several advantages including shareability, defining value sets per data field, and increasing note quality without increasing total charting times [51]. Smart templates can also support scalable and automated data extraction which reduces time and labor costs associated with manual chart data extraction.

### 4.2. Recommended Patient-Reported Outcome/Symptom Surveying Tools

Of the four validated multi-domain PROM symptom tools, EPIC-26 was prioritized given its ability to capture and track longitudinal, and comprehensive data related to urinary, sexual, bowel, systemic, and quality of life domains. The panel’s selection of the EPIC-26 tool to assess symptom profiles is in alignment with recommendations made by Martin et al. in 2015 who endorsed EPIC-26 as a standard set of PROMs for men with localized prostate cancer [52]. It is also commonly used in large studies, such as the ProtecT Study Group who valued its assessment of the four domains and QOL in PCa patients managed with surveillance, surgery or radiotherapy. However, its adoption in clinical practice is likely low as reported by the Expert Panel (24%), thereby highlighting an ongoing challenge in the ability of organizations/hospitals to address the barriers to longitudinal PRO monitoring implementation [53]. Similar approaches to develop smart forms for TRTs in electronic health record systems can be considered for recreating PROM tools in a structured manner. An alternative approach is the development and delivery of automated surveys to consenting patients through electronic data capture systems, such as REDCap^®^. The COVID-19 pandemic resulted in an upsurge of REDCap^®^ surveys being remotely distributed to patients to support COVID-19 testing and aggregate reporting [54]. These promising results present new opportunities to facilitate structured collection of PCa-specific PROMs across various healthcare settings.

### 4.3. Recommended Reporting of Disease Control and Response Metrics

Regarding disease control and response metrics, general definitions for survival (i.e., overall, cancer-specific, metastasis-free, and biochemical recurrence-free survival) in localized prostate cancer have been recommended by other consensus studies without explicit definitions on how to report data related to the extent of disease recurrence, if present [52]. To address this issue, there are continued efforts in specialties, such as radiology, to develop imaging reporting risk stratification systems to better inform clinicians of the disease status of patients after their anatomy has been altered post local therapy [55]. As reported in Table 3, our panel recommended further expansion of value sets related to anatomical sites involved in recurrence patterns (i.e., primary, pelvic nodal, and distant sites). From a comprehensive operational ontology standpoint, this recommendation can be translated into several DCM-based KDEs, such as primary_site_recurrence (i.e., with values including combinations of “none”, “prostate/bed”, or “seminal vesicles”) and nodal_site_recurrence (i.e., values including “none”, “pelvic”, “para-aortic”, and “distant”). Extent of non-nodal metastatic disease can also be classified as a numerical representation of osseous metastasis, with or without the presence of visceral lesions. These elements of metastatic disease have already been incorporated into trial-based criteria [41,42]. However, variations in the definitions of low- versus high-risk/volume metastatic disease present a challenge in comparing patient cohorts and the efficacy of evolving therapies (i.e., the “high volume” definition in CHAARTED includes four or more osseous metastases while LATITUDE defines “high risk” as having two or more factors that can include three or more osseous metastases). A common numerical threshold for defining oligometastatic/oligorecurrent PCa in recent randomized trials is the maximum presence of three to five involved sites (NCT04787744, NCT04115007, NCT03784755, NCT02759783, NCT03630666, NCT03569241, NCT04031378, NCT03940235, and NCT04037358). Therefore, in alignment with our panel consensus, we recommend that the extent of nodal and/or osseous lesions be reported in a numerical format in clinical records with the option of reaching a particular threshold of five lesions or more.

### 4.4. Study Limitations

Our Delphi study has inherent risks of bias including response bias (i.e., who decided to participate in this study), attrition bias (i.e., attrition in panelist participation during the second survey), and cognitive bias (i.e., framing and anchoring, and the bandwagon effect). We aimed to mitigate these biases by expanding the initial multidisciplinary invitation list, preserving anonymity of participants during structured group interactions, and asking panelists to share additional qualitative comments that could be used to support or oppose arguments for KDE selection. Despite such bias risks, the Delphi method is accepted as a useful technique for information collection and knowledge building on “informed opinion and subjective expert judgments as well as experienced-based interpretations” [38,56]. Another potential limitation includes our use of more relaxed definitions for reaching consensus (i.e., 50% or more). While lower thresholds of agreement are generally not seen in other Delphi studies, the authors felt it was appropriate for the overarching goal of this consensus procedure which was to provide multidisciplinary stakeholder input on KDEs to be included in our operational ontology for PCa. Additionally, the panel had a high representation of radiation oncologists compared to other specialties. This unbalanced distribution of specialists was intentional to optimize the O3 development phase and is counterbalanced by the allowance of more non-radiation oncology specialists on the panel due to the feasible, remote and electronic nature of the Delphi method. Lack of dynamic in-person conversations, variable panelist experiences with managing PCa patients, and restriction of Delphi rounds could have also affected the data elements that were brought up for review and ultimately included in the KDE list. Moreover, this study does not offer recommendations on radiation treatment doses to reduce the risk of TRTs as this was out of the current scope, and radiation schedules are being continuously investigated for efficacy and toxicity effects in several randomized clinical trials [50]. However, O3 currently supports the standardized collection of granular radiotherapy data (i.e., modality, daily dose, and fractionation) in addition to capturing patient and cancer characteristics, TRTs, DCMs, and PROMs as recommended by the BDSC and Expert Panel. This ensures that while treatments and technology evolve, comprehensive capture of core data elements using structured and routinely updated fields remains standardized across institutions. Overall, our work is synergistic with other ongoing efforts to define data elements in prostate cancer care for the construction of analytical pipelines and improved health information sharing [57].

### 4.5. Operational Ontology Build and Future Directions

Translation to Operational Ontology Build. Effective use and exchange of information in PCa care and research requires identification and standardization of data elements that are meaningful and comprehensive (i.e., pertaining to multiple domains of care including diagnosis, staging, toxicity reporting, and major clinical outcomes). The multidisciplinary Delphi consensus procedure resulted in a detailed report of KDEs related to TRTs, PROMs, and DCMs. Once provided with an endorsed KDE value set, the BDSC, in partnership with multiple external stakeholders, began the expansion of an operational ontology for oncology (O3) with the latest version available online and with links to download the complete set as a spreadsheet or in JavaScript Object Notation (JSON) format for electronic processing [58]. Figure 3 illustrates the O3 website for PCa-specific attributes including designated priority levels as recommended by the Expert Panel (i.e., red box for rectal hemorrhage listed as Priority 1). When feasible, O3 attributes, including those specific to PCa, were mapped to attributes found in other coding systems, such as SNOMED-CT to facilitate interoperability among those systems [59]. Technical details on the BDSC’s iterative process for developing O3 and affiliated consensus-driven information standards are reported by Mayo et al. [60].

To our knowledge, this is the first study of its kind to immediately translate expert-based recommendations from a Delphi study into technical efforts in operational ontology development. Future directions include the use of this existing Delphi-to-ontology protocol by the BDSC to update clinical guidelines and O3 with new knowledge and to promote the identification of KDEs and expansion of O3 related to other disease sites (i.e., head and neck, thoracic, etc). In clinical practice, implementation of recommended symptom monitoring tools and prospective KDE collection can be associated with several barriers including workflow challenges, survey fatigue and limitations in technology [61,62]. Therefore, implementation strategies and studies should be encouraged to ensure that PCa-specific KDEs are consistently captured with high data quality. Formative evaluations are especially helpful for clinical implementation endeavors as they are rigorous, multi-phase (i.e., developmental [pre-implementation interviews], implementation-focused, progress-focused, and interpretive [post-implementation surveys/interviews]) assessment processes that identify facilitators and barriers to change as well as potential solutions for optimizing the desired change [60,61]. Dissemination of successful strategies in various settings (i.e., academic centers, and private practice) would be ideal to facilitate KDE capture at a national level. O3 will also require routine updates by the BDSC, especially as it begins to be adopted by multidisciplinary cancer teams. The overarching goal is to see parallel adoption of prospective KDE collection and use of O3 over time in order to enable the construction and sharing of multi-institutional comprehensive, high-quality “real world” datasets that support robust clinical practice and research applications.

## 5. Conclusions

Through the application of a modified Delphi technique, a multidisciplinary team of prostate cancer experts was able to develop recommendations for a key dataset of treatment-related toxicities, patient-reported outcome metrics, and disease control metrics. These recommendations are ideal for all clinical practices and research and were successfully translated into building an endorsed, web-based operational ontology that can facilitate scalable and accurate information retrieval and exchange related to cancer care.

## Figures and Tables

**Figure 1 cancers-15-03121-f001:**
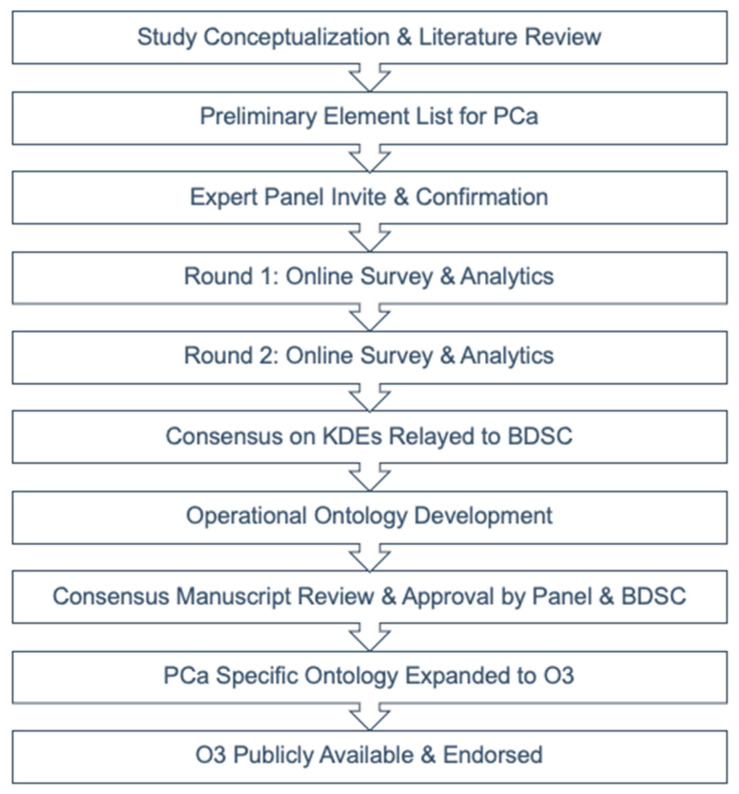
Delphi and operational ontology process flowchart.

**Figure 2 cancers-15-03121-f002:**
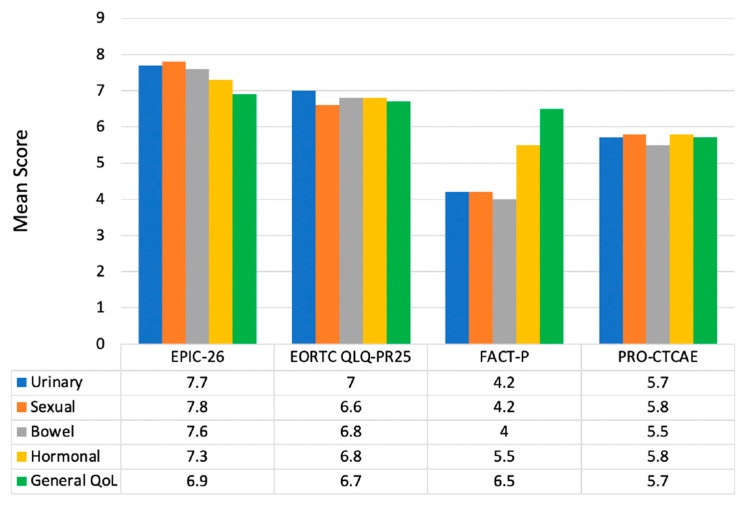
Mean scores of importance per subdomain of four validated symptom tools (higher score reflects higher priority/relevance).

**Figure 3 cancers-15-03121-f003:**
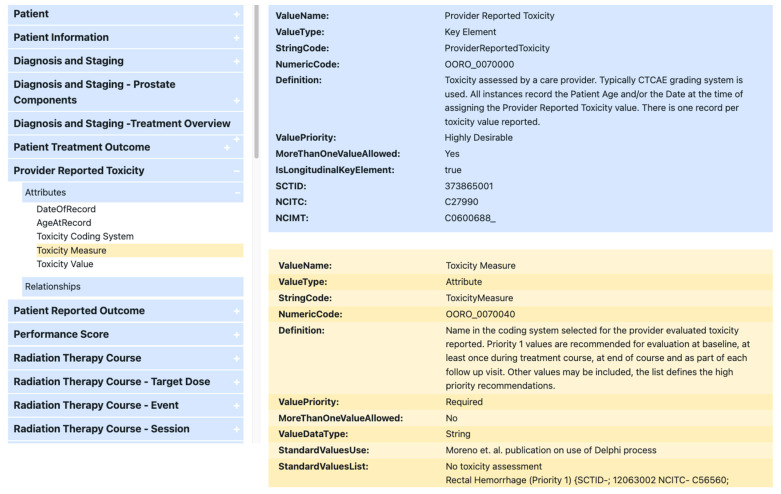
Operation ontology for oncology (O3) website for prostate cancer key data elements.

**Table 1 cancers-15-03121-t001:** Characteristics of the Delphi multidisciplinary Expert Panel (*n* = 39).

Panel Characteristics	Count (%) or Average
Age, mean (range), years	49.5 (34–70)
Gender	
Male	26 (67)
Female	13 (33)
Specialty	
Radiation Oncology	22 (56)
Radiation Physics	7 (18)
Medical Oncology	7 (18)
Urology	3 (8)
Practice Setting	
Academic	31 (79)
Private	1 (3)
Not answered	7 (18)
Approximate years in practice	17.7
Average patient caseload per week	30

**Table 2 cancers-15-03121-t002:** Treatment-related toxicity tier list categorization by the Expert Panel.

Tier 1 TRTs: Minimum KDE	Selected Tier Percentage	AI
Rectal Hemorrhage	96.4	0.93
Urinary Incontinence	92.9	0.86
Urinary Retention	92.6	0.86
Erectile Dysfunction	88.9	0.79
Hematuria	85.7	0.75
Dysuria	85.7	0.74
Rectal Fistula	77.8	0.64
Urinary Urgency	77.8	0.64
Urinary Frequency	77.8	0.63
Urinary Fistula	77.8	0.63
Proctitis	74.1	0.60
Fecal Incontinence	70.4	0.57
Diarrhea	70.4	0.53
Rectal Perforation	66.7	0.54
Rectal Ulcer	63.0	0.52
**Tier 2: Strongly Encouraged**		
Libido Decrease	66.7	0.49
Gynecomastia	66.7	0.48
Depression	59.3	0.43
Ejaculation Disorder	59.3	0.42
Rectal Fissure	55.6	0.43
Hemorrhoids	53.9	0.45
Rectal Mucositis	51.9	0.37
Rectal Stenosis	51.9	0.37
Fatigue	51.9	0.36
Hot Flashes	51.9	0.40
Rectal Pain	48.2	0.48
Cystitis (Non-infective)	48.2	0.41
Urinary Tract Pain	48.2	0.35
Bladder Spasms	48.2	0.34
Urinary Tract Obstruction	40.7	0.42
**Tier 3: Not Required**		
Superficial Fibrosis	96.2	0.92
Anorexia	88.5	0.79
Nausea	84.6	0.73
Peripheral Neuropathy	84.6	0.72
Dehydration	80.8	0.68
Radiation Dermatitis	80.8	0.66
Vomiting	80.8	0.66
Pelvic Infection	61.5	0.48
Anal Mucositis	57.7	0.49
Constipation	53.9	0.42
Prostatic Hemorrhage	50.0	0.39
Urinary Tract Infection	44.4	0.36

Abbreviations: AI, agreement index; KDE, key data elements; TRTs, treatment-related toxicities.

**Table 3 cancers-15-03121-t003:** Panel ranking of disease response assessment options.

Name	% Yes	AI *
Biochemical Recurrence	100%	1
Recurrence at Primary, Pelvic Nodal, and Distant Sites	92%	0.85
No evidence of disease (NED) or Complete response	88%	0.78
Progressive Disease	84%	0.72
Recurrence at Distant Site(s) Only	80%	0.67
Recurrence at Primary and Pelvic Nodal Sites	80%	0.67
Recurrence at Primary and Distant Sites	80%	0.67
Recurrence at Pelvic Nodal and Distant Sites	80%	0.67
Recurrence at Primary Site Only	76%	0.62
Recurrence at Pelvic Nodal Site(s) Only	76%	0.62
Under Treatment	72%	0.58
*Stable Disease*	52%	0.48
*Indeterminate (possible pseudo-progression)*	46%	0.48
*Partial Response*	40%	0.5

* AI, agreement index. Italicized metrics did not reach consensus threshold for inclusion.

## Data Availability

Data will be made available upon reasonable request to the corresponding author.

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
