# Peer review of "Identification of Key Elements in Prostate Cancer for Ontology Building via a Multidisciplinary Consensus Agreement"

_cancers, 2023, doi:10.3390/cancers15123121_

Round 1

Reviewer 1 Report

The study is an interesting one. I have only a few suggestions.

1. Research questions may be framed and answered in the discussion section.

2. A literature review with the SOTA studies in the domain may be incorporated.

3. Limitation of the study as a whole may be discussed.

4. Future research directions may be proposed.

5. Comparison of the results with the existing similar studies may be encouraged.

Author Response

Response to Review #1

The study is an interesting one. I have only a few suggestions.

  1. Research questions may be framed and answered in the discussion section. The authors appreciate this feedback and have re-organized and expanded the discussion section using the following sub-headers:

4.1 Recommended Treatment-Related Toxicity Reporting

4.2 Recommended Patient Reported Outcome/Symptom Surveying Tools

4.3 Recommended Reporting of Disease Control and Response Metrics

4.4 Study Limitations

4.5 Operational Ontology Build and Future Directions

  1. A literature review with the SOTA studies in the domain may be incorporated. We agree that a literature review of SOTA studies (and guidelines) in PCa is valuable to the study, and we would like to refer the reviewer back to the manuscript section titled ‘Survey Design and Administration’, lines 161-162 where we mention that ‘we performed a comprehensive literature review to develop a list of potentially important data elements related to PCa.’ To further explain, the BDSC spent several months performing iterative reviews of the literature in PubMed to identify abstracts and peer-reviewed studies with potentially relevant information related to our data elements of interest. However, this was not a formal systematic review as the domain was considered too extensive (i.e., PROMs and DCMs can be referenced in multiple ways and many studies describe them in PCa subpopulations only). Furthermore, the BDSC wanted to keep the initial survey flexible with the inclusion of more open-ended questions to stimulate new item generation, an approach commonly used in modified Delphi techniques.1 To address the reviewer’s comments, we have incorporated discussions of large PCa studies and/or guidelines in the discussion section (see item #1 above) and have revised the methods section with the following statement:

“We performed an informal but comprehensive literature review to develop a list of potentially important data elements related to PCa. Specifically, PubMed was iteratively queried using a variety of combinations of the following keywords and/or MeSH terms: prostate cancer, guidelines, treatment-related toxicities, adverse effects, disease control metrics, disease response assessment, prostate cancer surveillance, patient reported outcomes, symptom monitoring, PRO tools and/or validated questionnaires. Abstracts were reviewed for relevance and manuscript review was limited to publications related to large, PCa randomized clinical trials and/or PCa guidelines endorsed by nationally/internationally recognized organizations such as the American Urological Association (AUA), American Society for Radiation Oncology (ASTRO), European Society for Therapeutic Radiation and Oncology (ESTRO), and the European Society of Urogenital Radiology (ESUR).”

  1. Limitation of the study as a whole may be discussed. Several limitations can impact this study as a whole. To address this revision suggestion, the authors have considerably expanded the discussion paragraph on limitations to reflect risks of various types of biases, our approach to mitigate such biases, and other limitations such as restriction of Delphi rounds and lack of dynamic in-person conversations. 
  1. Future research directions may be proposed. The authors fully agree with this recommendation and have added a new paragraph to the end of the manuscript focused on future directions. Several new references have also been added to discuss challenges on implementing the panel’s recommendations and how formative evaluation methods can be useful to study and improve clinical implementation efforts. 
  1. Comparison of the results with the existing similar studies may be encouraged. The authors acknowledge that this is important and have expanded the discussion section as previously mentioned. However, there is no comparative Delphi study for supporting the build of operational ontologies. We believe this is a major advantage of our study with publication benefits including the reporting of our methodology for future applications by the BDSC or other groups interested in clinical data standardization.

Reviewer 2 Report

In the article entitled “Identification of key elements in prostate cancer for ontology 2 build: a multidisciplinary consensus agreement”, the authors aimed at formulating an expert panel-based consensus on PCa-specific key data elements. They used the Delphi method and PCa experts developed a two-tiered 30-item list of treatment-related toxicities for standardized clinical data capture. At the end of the study, their findings have been used to develop a comprehensive operational ontology for PCa care in order to facilitate knowledge sharing and scalable machine-learning approaches.

Although the aim of this manuscript is very amazing, I think it is subjected to a big bias, that is the subjectivity of the protocol.

This study could represent a useful tool in the future  but it needs to be better standardized and based on objective and discrete parameters far from multiple levels of questions.

Are there any evidences for the analysis conducted by the authors by validating them through retrospective studies ar datasets?

The authors should add an ABBREVIATIONS Section to the manuscript in order to clarify the meaning of such acronyms reported within the text (ie. BDSC, TRT, PROM, DCM, etc.

In conclusion, I think the manuscript cannot be taken in consideration for publication in its present form.

The text needs a moderate editing of English language.

Author Response

Response to Reviewer #2:

In the article entitled “Identification of key elements in prostate cancer for ontology 2 build: a multidisciplinary consensus agreement”, the authors aimed at formulating an expert panel-based consensus on PCa-specific key data elements. They used the Delphi method and PCa experts developed a two-tiered 30-item list of treatment-related toxicities for standardized clinical data capture. At the end of the study, their findings have been used to develop a comprehensive operational ontology for PCa care in order to facilitate knowledge sharing and scalable machine-learning approaches.

Although the aim of this manuscript is very amazing, I think it is subjected to a big bias, that is the subjectivity of the protocol. The authors fully agree with the reviewer’s comments and have made substantial revisions to the limitations section of the discussion. Overall, it is known that Delphi studies are subject to big bias; however, their strength primarily lies in information collection and a structured discussion approach for consensus formation from a group of experts in a particular domain. The purpose of this study was to use this collected information and knowledge to build a robust operational ontology and a clinical framework for guiding clinicians and researchers on key data elements to prospectively capture during PCa care. The limitations paragraph now includes the following statement:

“Our Delphi study has inherent risks of bias including response bias (i.e., who decided to participate in this study), attrition bias (i.e., attrition in panelist participation during the second survey), and cognitive bias (i.e., framing and anchoring, and the bandwagon effect). We aimed to mitigate these biases by expanding the initial multidisciplinary invitation list, preserving anonymity of participants during structured group interactions, and asking panelists to share additional qualitative comments that could be used to support or oppose arguments for KDE selection. Despite such bias risks, the Delphi method is accepted as a useful technique for information collection and knowledge building on “informed opinion and subjective expert judgements as well as experienced-based interpretations” [1,2].”

This study could represent a useful tool in the future, but it needs to be better standardized and based on objective and discrete parameters far from multiple levels of questions. The authors appreciate this comment and agree that data standardization should include both objective and subjective metrics. Most objective and discrete parameters (i.e., dose fractionation, receipt of surgery, age at diagnosis) are relatively well-known and defined such as in the AAPM’s Task Group 263 Report for ‘Standardizing Nomenclatures in Radiation Oncology’. This Delphi study did not seek to reconfirm those parameters but instead tackle various PCa concepts that have considerable ambiguity (i.e., what is the definition of DCMs?) or practice patterns (i.e., reporting of TRTs in clinical practice). The OORO website which is included in the manuscript can guide clinicians and physicists on how to explicitly report such important objective and discrete parameters in addition to new ones designed from the Expert Panel recommendations and related to TRTs, PROMs, and DCMs.

Are there any evidences for the analysis conducted by the authors by validating them through retrospective studies or datasets? As this is the first Delphi consensus-to-operational ontology study to date, there are no other existing studies to validate our approach. However, the methods we used for forming consensus on PCa-specific KDEs were primarily founded on Delphi methodology articles[3–6] and through continued consultation with our biostatistician, Dr. Lin. The AAPM BDSC[7] has extensive experience in developing data standards and so the translation to an operational ontology was based on their expertise in this field. Future studies, including the testing of OROO and the adoption of KDE collection in clinical practice, are needed for validation of their utility in building robust datasets with preserved semantics.

The authors should add an ABBREVIATIONS Section to the manuscript in order to clarify the meaning of such acronyms reported within the text (ie. BDSC, TRT, PROM, DCM, etc.). The authors appreciate this feedback and have added a section at the end of the manuscript, labeled “6. Abbreviations”.

Additional Revisions to the Manuscript

When appropriate, the English language has been modified in the manuscript in response to comments lefts by Reviewers 2 and 3.

References

  1. Winkler, J.; Moser, R. Biases in Future-Oriented Delphi Studies: A Cognitive Perspective. Technol Forecast Soc Change 2016, 105, 63–76, doi:10.1016/J.TECHFORE.2016.01.021.
  2. Beiderbeck, D.; Frevel, N.; Gracht, H.A. von der; Schmidt, S.L.; Schweitzer, V.M. Preparing, Conducting, and Analyzing Delphi Surveys: Cross-Disciplinary Practices, New Directions, and Advancements. MethodsX 2021, 8, 101401, doi:10.1016/J.MEX.2021.101401.
  3. Delphi Method | RAND Available online: https://www.rand.org/topics/delphi-method.html (accessed on 1 August 2021).
  4. Ogbeifun, E.; Agwa-Ejon The Delphi Technique: A Credible Research Methodology.
  5. Hasson, F.; Keeney, S.; McKenna, H. Research Guidelines for the Delphi Survey Technique. J Adv Nurs 2000, 32, 1008–1015, doi:10.1046/j.1365-2648.2000.t01-1-01567.x.
  6. Avella, J.R. Delphi Panels: Research Design, Procedures, Advantages, and Challenges. International Journal of Doctoral Studies 2016, 11, 305–321.
  7. AAPM Big Data Subcommittee (BDSC) Available online: https://aapmbdsc.azurewebsites.net/Home/About?ReturnUrl=%2F (accessed on 2 August 2021).

Reviewer 3 Report

The article provided by Moreno and colleagues is of interest in the cancer field and in particular in PCa. However, in my opinion it needs some improvements concerning the usability of the related findings in the article. Below the main points that could be useful to improve the quality of the paper: -In my opinion, the authors should make an effort to better disclose the use of the findings in the paper and how the findings can be useful for physicians. Probably reporting for example some case studies to focus the usage of the findings can be important. -authors reported treatment-related toxicities, no mention about the kind of treatments doses and so on is reported and how the model can help the physicians to overcome this kind of toxicities. -it can be interesting to report the fact that this kind of approach can be updated with novel acquire knowledge and how can be implemented in the future. After addressing these point the paper can be reevaluated. minor row 77 reference is not in the correct format

English language required some revisions that can be addressed in the revised version.

Author Response

Response to Reviewer #3:

The article provided by Moreno and colleagues is of interest in the cancer field and in particular in PCa. However, in my opinion it needs some improvements concerning the usability of the related findings in the article. Below the main points that could be useful to improve the quality of the paper:

In my opinion, the authors should make an effort to better disclose the use of the findings in the paper and how the findings can be useful for physicians. Probably reporting for example some case studies to focus the usage of the findings can be important. The authors agree with this opinion and have made considerable revisions in the discussion section of the manuscript. These revisions include further specification on how Expert Panel recommendations are in alignment with existing publications and our considerations on how to implement these recommendations for consistent TRT, PROM, and DCM KDE collection (i.e., use of smart forms or problem-oriented templates; use of formative evaluation methods in implementation studies).

Authors reported treatment-related toxicities, no mention about the kind of treatments doses and so on is reported and how the model can help the physicians to overcome this kind of toxicities. The authors specifically chose to exclude panel review of and recommendations on radiation treatment doses as this was out of the scope of the study (i.e., not part of TRTs, DCMs, or PROMs). Data elements related to radiation therapy such as daily radiation dose (cGy), total number of fractions, and the definition of clinical target volumes or CTVs have all been previously defined in the ‘Standardizing Nomenclatures in Radiation Oncology’ Task Group 263 Report published by AAPM.[1] The authors agree that the incidence and severity of treatment-related toxicities can be dose-dependent. However, given continuously evolving radiotherapy techniques, including the recent exploration of hypofractionated or stereotactic body radiotherapy regimens for PCa, recommendations on specific treatment doses are strongly encouraged to be derived from randomized trials investigating the efficacy and toxicity-effect of certain radiation schedules such as in the PACE-B trial.[1] The operational ontology developed by AAPM BDSC supports standardized reporting of granular radiation therapy details and also includes structured fields for TRT reporting based on reviewed and recommended symptom survey tools such as EPIC-26 and IPSS, both of which were used in the PACE-B trial. To address this concern, the authors have revised the discussion section to include the following statement:

“Moreover, this study does not offer recommendations on radiation treatment doses to reduce the risk of TRTs as this was out of the scope of the study, and radiation schedules are being continuously investigated for efficacy and toxicity effects in randomized clinical trials[2]. However, OORO currently supports the standardized collection of granular radiotherapy data (i.e., modality, daily dose, and fractionation) in addition to capture of patient and cancer characteristics, TRTs, DCMs, and PROMs as recommended by the BDSC and Expert Panel. This ensures that while treatments and technology evolve, comprehensive capture of core data elements using structured and routinely updated fields remains standardized across institutions.”

It can be interesting to report the fact that this kind of approach can be updated with novel acquire knowledge and how can be implemented in the future. The authors appreciate this comment and agree that this is a valuable advantage to the Delphi-to-ontology template we have developed in this study. The authors would like to guide the reviewer to a newly developed ‘future directions’ paragraph in the revised manuscript at the end of the discussion section which includes the following statement:

“Future directions include the use of this existing Delphi-to-ontology protocol by the BDSC to update clinical guidelines and OROO with new knowledge and to promote the identification of KDEs and expansion of OORO related to other disease sites (i.e., head and neck).”

Minor row 77 reference is not in the correct format. The authors have corrected this by removing the superscript number (the reference is in proper format for the following sentence).

Additional Revisions to the Manuscript

  1. When appropriate, the English language has been modified in the manuscript in response to comments lefts by Reviewers 2 and 3.

References

  1. Task Group, A. Standardizing Nomenclatures in Radiation Oncology The Report of AAPM Task Group 263. 2018.
  2. Tree, A.C.; Ostler, P.; van der Voet, H.; Chu, W.; Loblaw, A.; Ford, D.; Tolan, S.; Jain, S.; Martin, A.; Staffurth, J.; et al. Intensity-Modulated Radiotherapy versus Stereotactic Body Radiotherapy for Prostate Cancer (PACE-B): 2-Year Toxicity Results from an Open-Label, Randomised, Phase 3, Non-Inferiority Trial. Lancet Oncol 2022, 23, 1308–1320, doi:10.1016/S1470-2045(22)00517-4.

Round 2

Reviewer 2 Report

In the 2nd version of the article entitled “Identification of key elements in prostate cancer for ontology build: a multidisciplinary consensus agreement”, the authors responded to my comments.

With respect to my comments, they added some data to the manuscript.

My consideration is that this kind of analysis requires an experimental validation first, ALWAYS. Despite these considerations, I consider that the manuscript is clear to read and this study could represent a useful reference tool.

I think it would be important to add some experimental models and physical samples (clinicians or cell cultures) to support their analyses and conclusions.

No comments

Reviewer 3 Report

Authors addressed my main concerns

Minor editing of English language required